# OpenReview forum: "IterGen: Iterative Semantic-aware Structured LLM Generation with Backtracking"
_ICLR.cc/2025/Conference — ICLR 2025 Poster_

### Official Review · Reviewer_My2d · 2024-11-02

**Soundness:** 3
**Presentation:** 3
**Contribution:** 2
**Rating:** 6
**Confidence:** 5

**Summary:**

- This paper proposes a system for grammar-constrained generation from LLMs called IterGen. Relative to existing approaches to this problem, the main contribution of this work is to support backtracking to previous intermediate states, allowing for correction of mistakes mid-generation.
- At a technical level, this requires solving several key challenges: (1) mapping between the CFG symbol space and the LLM vocabulary (referred to in the literature as the *token misalignment* problem), (2) maintaining traces of the decoding history to allow for backtracking, and (3) caching Transformer KV states to enable decoding to be efficiently resumed from intermediate points.
- Experiments are presented on two domains: SQL Generation on the Spider dataset (Yu et al., 2018) and an email privacy leakage task based on the Enron email dataset included in the DecodingTrust benchmark (Wang et al., 2023). IterGen is compared against a recent non-backtracking CFG-constrained generation system called SynCode (Ugare et al., 2024) as well as unconstrained decoding (termed STANDARD here). The three decoding methods are applied to small LLMs (≤3B) in the Quen2.5 and Llama-3.2 families. On Spider, IterGen is shown to provide improvements over unconstrained generation and modest improvements over SynCode. On the privacy leakage task, IterGen is only compared to unconstrained generation and, by construction, reduces the number of email leaks to 0 while slightly increasing perplexity.

**Strengths:**

- Grammar-constrained generation is one of the rare research problems that comes with both thorny technical challenges and important real-world applications. Consider the problem of semantic parsing natural language queries into SQL, which is now being faced by virtually every enterprise company that is now attempting to deploy LLMs for business applications. For this reason, I view work any in this space as being of potentially high significance.
- Related to the above, Spider / SQL generation is a good choice for an evaluation domain as it showcases the application potential of this work. (Though as discussed below, I think this work would benefit from evaluation on another established benchmark besides Spider.)
- From a software systems standpoint, IterGen is clearly described. Even without seeing the code implementation, I feel confident that I could reimplement the core method.
- The related work (Sec. 5) is thorough and includes most of the main comparable works on grammar-constrained LM generation.

**Weaknesses:**

**Originality of research contributions:** Broadly speaking, IterGen reimplements existing solutions to technical challenges described in prior work as opposed to introducing new solutions. For instance, a variety of existing approaches address the token misalignment problem (e.g., Shin et al., 2021; Poesia et al., 2022; Willard and Louf, 2023; Moskal et al., 2024; Ugare et al., 2024). While token misalignment is discussed as a technical challenge throughout, as far as I can tell, IterGen is built on top of the SynCode library and uses the same solution to this problem described in Ugare et al., 2024. Similarly, other key challenges like maintaining trees for the decoding history and KV caching are (A) primarily engineering-related and (B) leverage existing engineering solutions built into libraries like HuggingFace. While there may be real practical value in combining these three technical solutions into one package, I’m not sure that this qualifies as a novel research contribution.

**Technical report vs. research paper:** Related to the above, while the paper reflects a good deal of engineering effort went into building IterGen, as written, this submission reads more like a systems paper / technical report than an ICLR paper. Significant portions of space are dedicated to describing the IterGen API from a user/developer perspective (e.g., Sec. 3.1). Additionally, the focus on evaluation metrics like execution time are features of a technical report that don’t necessarily count as a research contribution.

**Triviality of the algorithm in practice:** The authors position IterGen’s main contribution as the ability to navigate forward/backward through the grammar state space. However, in practical use cases, IterGen appears to reduce to rejection sampling. For instance, consider the privacy leakage evaluation presented in Sec. 4.2: algorithmically (Fig. 4), IterGen simply resamples generations from the LM until the generation does not contain the target email. Aside from the convenience of maintaining a KV cache for the prompt, it is unclear what benefits IterGen provides in this task over simply rejection sampling with the underlying LM.

**Missing rejection sampling baselines:** I am also concerned that a rejection sampling baseline is not reported for either evaluation. Especially given the way that the sampling algorithm is defined for the privacy domain (Fig. 4), I expect that IterGen would perform comparably to a rejection sampler (i.e., pass@k) with a matched sample budget. I suggest that the authors add an evaluation comparing IterGen against a sample-matched rejection sampling baseline.

**Evaluation tasks lack complexity:** Another way of viewing the above critique would be to say that the tasks chosen as evaluation domains are not complex enough to really showcase the benefits of the system.

- **Privacy leakage task:** This is especially true in the case of the email privacy leakage evaluation, where the ability to perform backtracking search is overkill for a domain where solutions are trees of depth=1. It seems like a critical revision would be to modify the privacy leakage task to actually require sequential decision-making. For instance, generating a sequence of employees forming a valid email chain (according to some underlying graph structure) while ensuring that email addresses remain private.
- **Spider:** Meanwhile, on Spider, it’s plausible that backtracking would be useful, but the evaluation metrics don’t provide enough information to conclude whether backtracking is what is responsible for the observed improvements in performance (see Questions 1-2 below).
- **Additional domains:** Especially since the privacy leakage domain is a toy task defined by the authors, I would really like to see evaluation on another public benchmark; e.g., BenchCLAMP (Pauls et al., 2022), SMCalFlow (Andreas et al., 2020), the NLV corpus (Srinivasan et al., 2021), or similar.

**Lack of empirical comparison to related work:** While the related work section is thorough and does mention most of the main comparable works on grammar-constrained LM generation, it is notable that none of these methods were benchmarked as baselines in the evaluations. This is especially a problem as the privacy leakage domain is a toy task, so Spider is the only point of empirical comparison between IterGen and other published work.

- re: “Synchromesh’s code is not publicly available” (L519) — While this is technically true, there does exist a popular open-source reimplementation from one of the original authors (https://github.com/kanishkg/synchromesh). For this reason, it feels like a bit of a manufactured excuse to say that Synchromesh—or any of the other related works, for that matter—cannot be compared to in the evaluations. I suggest that the authors evaluate against the open-source Synchromesh implementation and report the results for the rebuttal.

**Framing of semantic constraints as a contribution:** The ability to enforce various semantic properties beyond syntax is framed as one of the key contributions of IterGen (e.g., L42-48). However, I disagree with the notion that other tools “cannot apply semantic constraints to the generation process” (L486). This feels like an unfair mischaracterization, seeing as the application of semantic constraints in this paper is limited to checking string equality. In other words, the examples of semantic constraints demonstrated in this paper are quite trivial, and it is easy to implement a similar string-matching process using a comparable library like Guidance or Outlines. Thus the claim that semantic constraints are part of the unique contribution story of this paper feels hollow to me.

**Technical limitations:** Section 6 mentions that IterGen does not supported batched generation, which is something that other existing works do support (e.g., Lew et al., 2023 support asynchronous execution with batching as well as KV caching). I’m sympathetic to the unique challenges introduced by backtracking, but I think this is an important technical limitation to resolve if IterGen is going to be of practical value as a software tool.

## Other nits

- In Sec. 4.1, various numbers are cited when comparing IterGen against the other methods:

    > ITERGEN improves over both baselines with an average overall accuracy of 41.6% and an execution percentage of 75.1%, compared to 27.6% accuracy and 47.4% execution rate for STANDARD generation. It outperforms SYNCODE, which has an average accuracy of 35.1% and an execution rate of 63.4%.
    >

    None of these numbers appear in Table 1 — is this because these are averages over the entire column? If so, I suggest that the authors include a row labeled AVERAGE in Table 1 to make explicit that these numbers are being aggregated over multiple different LLMs of different families and parameter counts.

- $\Sigma$ is first defined over characters (Line 88) and then in the next section it is defined over LM tokens (Line 94). Given that a key issue in this space is *character-token misalignment*, it seems like an oversight to elide this distinction in the terminology.
- SQL generation program is nice for its simplicity but it feels like more could be done with the grammar to ensure validity beyond just checking valid column and table names (theoretically you could just substitute a grammar that hardcodes these, though it would be very large)

**Questions:**

1. In Table 1, do the Tokens and Time columns include backtracking? If so, how is it possible that IterGen in many cases generates fewer tokens than SynCode and Standard decoding? Does that mean that in practice, backtracking rarely occurs?
2. How often does IterGen get stuck? It seems like this method is prone to getting stuck due to using greedy decoding + backtracking search. I suggest that the authors update Table 1 to include the number of forward and backward calls made by IterGen as well as the number of times that the maximum number of iterations was reached.
3. “We experiment with a range of state-of-the-art LLMs under 3B parameters?” What is the motivation for this model size restriction? Is there a technical reason why larger-scale LMs were not evaluated?
4. I am confused about the recurrence penalty, specifically “$\gamma=1$ is equivalent to no penalty” (A.3, L746): Is this a typo? The rest of this paragraph implies that higher values correspond to higher penalties. (If this is not a typo, then I suggest changing the wording from “recurrence penalty” to something with the opposite polarity, such as “recurrence tolerance”.)

---

### Official Review · Reviewer_32H8 · 2024-11-03

**Soundness:** 4
**Presentation:** 4
**Contribution:** 4
**Rating:** 8
**Confidence:** 4

**Summary:**

The authors present a programmatic interface to a grammar-constraint generation system.  Unlike prior approaches, they provide a mechanism to insert backtracking/resampling steps in the generation process in a way that leverages the hierarchical structure of the grammar.  For example, if one seeks to generate an SQL statement (as they do in their experiments), they can regenerate the column name and table names until a valid one is found.  This is also helpful for non-context-free well-formedness checks, like ensuring that variable names are defined before use.  The ability to do non-context-free checks seems like a big advantage over other papers focusing only on context-free grammars.  The authors show strong empirical results on two tasks.

**Strengths:**

The authors present a programmatic interface to a grammar-constraint generation system.  Unlike prior approaches, they provide a mechanism to insert backtracking/resampling steps in the generation process in a way that leverages the hierarchical structure of the grammar.  For example, if one seeks to generate an SQL statement (as they do in their experiments), they can regenerate the column name and table names until a valid one is found.  This is also helpful for non-context-free well-formedness checks, like ensuring that variable names are defined before use.  The ability to do non-context-free checks seems like a big advantage over other papers focusing only on context-free grammars.  The authors show strong empirical results on two tasks.

**Weaknesses:**

The programmatic interface seems to be pretty tied to incremental parsing (e.g., LR), which has limitations in the grammar formalisms that it can correctly parse.  How might ambiguous grammars be supported?  For example, could these methods be broadened to Earley parsing?  Please add some discussion of the limitations of the supported grammar formalism to the paper for camera-ready.

**Questions:**

The proposed programmatic interface (`forward`, `backward`, and `view`) resembles the kinds of callbacks available in parsing libraries (like Lark in Python).  Could the interface be made similar to the existing callback interface?

A big part of the contribution of this paper is a new API for users of LLM technologies.  Will the code be released?  Can you share any words about the tool's usability (e.g., the learning curve of using the tool for a new task and the difficulty of debugging)?  Are any debugging strategies/interfaces supported by the tool?

Can you compare/contrast your proposed interface with that of `guidance`?  Both provide a mixed Python--grammar interface.

---

### Official Review · Reviewer_D1RM · 2024-11-04

**Soundness:** 2
**Presentation:** 2
**Contribution:** 2
**Rating:** 6
**Confidence:** 2

**Summary:**

This paper proposes IterGen, a constrained decoding algorithm that allows both forward and backward in grammar-constrained generation. Backward allows the model to walk back when the (partial) generation is not semantically correct, reducing the semantic errors in LLM generation.

IterGen provides some primitives that operate on the partial syntax tree of the current generation, with which one can write constrained decoding programs more expressively.

The authors demonstrated IterGen's effectiveness on two tasks:
 1. text to SQL, where the previous constrained decoding method SynCode can generate incorrect column names that violate the schema.
 2. Email generation, where the previous decoding method can generate private email accounts that are leaked in the training set.

**Strengths:**

1. The algorithm is well-motivated and clearly described. It provides a better abstraction for people to create decoding constraints that go beyond context-free grammar.
2. IterGen shows significant improvements over the baselines in the two tasks evaluated when the model under test is small, including accuracy, execution, and time / token taken.

**Weaknesses:**

### 1. The two tasks can be solved with forward-only constrained decoding.

Since the valid column names and leaked email addresses are known prior to the generation, the validity of column names and emails can be guaranteed with forward-only constraints. For example, when the column names only include `date` and `time` and the model is about to generate a column name, masking out all next tokens that are not the prefixes of these two strings will guarantee semantic correctness, as [1] did in guaranteeing that LLMs only generate correct, existing function names.

I would be more convinced of the merit of IterGen if the author could show:

 1) there is an abundance of tasks that only IterGen can solve, or
 2) on the two cases studied in the paper, IterGen performs better than forward-only decoding algorithms (that have the constraints I just described).

### 2. The gains from IterGen are insignificant on more capable models.

As Table 1 shows, the performance gain from IterGen is less significant when the model is larger or when the baseline performance is better, especially for Llama-3.2-3B, where most increases are <1%.

This makes me wonder whether the problem IterGen is trying to solve will exist for a longer time in the future, when we have models that are better trained.

I would be more convinced if the authors could show experiments on larger models.

### 3. Concerns about IterGen's efficiency.

While Table 1 suggests that IterGen uses fewer tokens and less time on average, I'm curious about IterGen's performance on individual cases, because it is not intuitive to me why IterGen is more efficient.

Compared to the baselines --- non-constrained generation and grammar-constrained forward-only generation --- IterGen needs extra tokens to go back and regenerate the semantically incorrect parts, which seems to need a larger time and token budget. I wonder for how many task instances is this the actual case and why on average IterGen does not use more tokens.

### Reference

[1] https://arxiv.org/abs/2310.07075

**Questions:**

Please see the weakness section.

---

> ### Comment · Reviewer_D1RM · 2024-11-25
>
> Thank you for the response.
>
> My second and third concerns have been addressed well, so I'm raising the score to 5.
>
> However, I'm not fully convinced by your answer to my first concern.
>
> If I understood correctly, to make sure the model only generates a correct column name, you need only to add the column names to a trie-like structure, which should take linear time with respect to the sum of name lengths. Could you further explain why it's taking so long?

---

> ### Author Response · Authors · 2024-11-25
>
> > If I understood correctly, to make sure the model only generates a correct column name, you need only to add the column names to a trie-like structure, which should take linear time with respect to the sum of name lengths. Could you further explain why it’s taking so long?
>
> We appreciate the reviewer’s prompt response.
>
> We hope the reviewer agrees that for the Vega-lite and privacy case studies, a purely grammar-based solution is infeasible. Below, we provide additional details to clarify why grammar-constrained generation tools often precompute token masks before inference for each grammar and thus constructing a new grammar for each schema is inefficient.
>
> Grammar-constrained generation requires precise knowledge of the parser state to identify valid tokens. For instance, in a partial generation like `SELECT stuid FROM student WHERE`, column names can follow, but other terminals are also valid, such as keyword (exists, case) or literals (strings, numbers). According to the SQL grammar, there are 18 terminal types possible in this context. Note that there are many such parser contexts where column name is valid, and in all those cases there will be a distinct set of other terminals that are valid. Identifying prefixes for these valid terminals requires iterating over the entire vocabulary, which becomes computationally expensive with large vocabularies. For example, Llamma models have 128k tokens, requiring up to 128k regex matches in this scenario.
>
> Additionally, within a single terminal, the generation state must be tracked. For example, even when generation is at `SELECT stuid FROM student WHERE ex`, the tool should filter out all syntactically valid tokens at this given point. Thus, this mapping depends on both the parser state and the state of the DFA for individual terminals, mapped to the set of valid tokens. Constructing this mapping involves iterating over all possible states and tokens offline, a process that is computationally intensive but essential for efficient constrained generation. To address this, SOTA tools precompute parser state-to-token mappings offline, which enables fast token filtering during inference (see [1], [2], [3]). As the reviewer suggested, a trie can also be utilized in this step, as demonstrated in [4]. However, this approach involves constructing a separate trie for each parser and automaton state, which remains comparably time-intensive.
>
> For SQL specifically, there are 80 terminal types, and each terminal’s DFA can have between 10 and 100 states. Constructing the mapping requires iterating over the entire vocabulary of 128k tokens for each state to determine which tokens are valid. This process of filtering valid tokens for each state is computationally expensive. On our machine, building this mapping for SQL takes approximately 5 minutes, and every time if the grammar is updated then the mapping needs to be constructed again.
>
> [1] XGRAMMAR: Flexible and Efficient Structured Generation Engine for Large Language Models (https://arxiv.org/pdf/2411.15100)
>
> [2] SynCode: LLM Generation with Grammar Augmentation (https://arxiv.org/abs/2403.01632)
>
> [3] Efficient Guided Generation for Large Language Models (https://arxiv.org/abs/2307.09702)
>
> [4] Guiding LLMs The Right Way: Fast, Non-Invasive Constrained Generation (https://arxiv.org/pdf/2403.06988)

---

> > ### Author Response · Authors · 2024-11-27
> >
> > > If I understood correctly, to make sure the model only generates a correct column name, you need only to add the column names to a trie-like structure, which should take linear time with respect to the sum of name lengths. Could you further explain why it's taking so long?
> >
> > Thank you again for your feedback.
> >
> > We wanted to check if the reviewer's question regarding precomputation time in grammar-constrained generation has been clarified. We also wanted to add that, in addition to the previous clarification, the approach in [1] uses a "regular language" to follow a specific tool syntax, which is why it requires only a DFA for constrained generation. In contrast, our work and the other works mentioned in previous answer consider "context-free languages" (e.g, SQL, Vega-Lite), which requires more expensive precomputation.
> >
> > We'd be happy to address any further questions the reviewer may have.
> >
> > [1] https://arxiv.org/abs/2310.07075 Don't Fine-Tune, Decode: Syntax Error-Free Tool Use via Constrained Decoding

---

> > > ### Comment · Reviewer_D1RM · 2024-11-27
> > >
> > > Thank you! It's clarified and I'm raising the score to 6.

---

### Comment · Area_Chair_PoTN · 2024-11-25
**Action Required: Respond to Author Rebuttals - Nov 27**

Dear ICLR Reviewers,

The author discussion phase is ending soon. Please promptly review and respond to author rebuttals for your assigned papers. Your engagement is critical for the decision-making process.

Deadlines:
- November 26: Last day for reviewers to ask questions to authors.
- November 27: Last day for authors to respond to reviewers.
- November 28 - December 10: Reviewer and area chair discussion phase.

Thank you for your timely attention to this matter.

---

### Meta-Review · Area_Chair_PoTN · 2024-12-21

**Metareview:**

The paper presents IterGen, a framework for iterative, grammar-guided LLM generation that enables both forward and backward navigation during the generation process through grammar symbols. The framework leverages symbol-to-position mapping and KV cache state management to enable structured generation with mid-process corrections.

Reviewers acknowledge several strengths of the work, including a well-motivated and clearly described algorithm, strong empirical results on smaller models, and potential practical value for real-world applications like SQL generation. The presentation is clear and the related work coverage is thorough. However, reviewers raise concerns about the paper's contributions. First, the technical novelty appears limited, as many of the core challenges (token misalignment, KV caching) are addressed using existing solutions from prior work. Second, while the system shows improvements over baselines, there are questions about whether simpler approaches like rejection sampling could achieve similar results, particularly given the relative simplicity of the evaluation tasks. The authors have addressed some concerns through additional experiments and clarifications, particularly around rejection sampling baselines and evaluations on more complex tasks like VegaLite.

Overall, I recommend acceptance given these considerations and the paper's clear practical value.

**Additional Comments On Reviewer Discussion:**

Mentioned in Meta review.

---

> ### Public Comment · ~Shubham_Ugare1 · 2025-02-24
>
> Dear AC,
>
>
> The reviewer discussion revealed a potential confusion regarding the scope of our work. To clarify this and to be more specific about the scope of our work we plan to update the title of our paper to "IterGen: Iterative Semantic-aware Structured LLM Generation with backtracking". We wanted to confirm that this change is acceptable.

---

### Decision · Program_Chairs · 2025-01-22

Accept (Poster)